# Assessing and Considering the Wider Impacts of Sport-Tourism Events: A Research Agenda Review of Sustainability and Strategic Planning Elements

**Ana Chersulich Tomino** [1,*], **Marko Perić** [1] 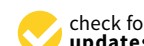 **and Nicholas Wise** [2]

1    University of Rijeka, Faculty of Tourism and Hospitality Management, Primorska 42, 51410 Opatija, Croatia; markop@fthm.hr
2    Liverpool Business School, Liverpool John Moores University, 4/6 Rodney Street, Liverpool L1 2TZ, UK; n.a.wise@ljmu.ac.uk
*    Correspondence: kersulica@gmail.com; Tel.: +385-(0)98-184-9174

**Abstract:** Sport-tourism events create a broad spectrum of impacts on and for host communities. However, sustainable sport-tourism events, which emphasize positive impacts, and minimize negative impacts, do not arise by chance—they need careful planning and implementation. This paper aims to review and systematize a wide spectrum of social impacts that outdoor sport-tourism events create from the perspective of key stakeholders and addresses strategic planning elements necessary for achieving event sustainability. To reach its objectives, the authors examined the Web of Science Core Collection (WoSCC) database, searching for relevant scientific papers focusing primarily on the impacts and legacy of sport events, strategic planning elements, and attributes necessary for achieving sustainability through a systematic quantitative review and content analysis. The results indicate that the relevant literature mostly focuses on economic impacts, followed by social and environmental impacts. Most studies focus on Europe and Asia, with the Olympic Games and FIFA World Cups being the most popular type of event studied. To systemize event and destination strategic elements and attributes for achieving sustainability, this study considers eight categories: social, cultural, organizational, logistic, communication, economic, tourism, and environmental. This paper identifies the main research gaps, proposes a new holistic sport-tourism events research agenda and provides recommendations so that organizers can avoid planning, organizing, financing mistakes and better leverage future sporting events.

**Keywords:** sport-tourism events; event impacts; outdoor events; strategic planning; legacy; organization; sustainability; systematic literature review; research agenda

## 1. Introduction

Over the past few decades, event tourism (culture, sporting and business-related events) has become a rapidly expanding segment of the leisure travel market [1–3]. The connection between sport and tourism is not new, and scholars have considered the rise of sporting events as one of the most significant components of event tourism and one of the most extensive elements of sporting tourism [4,5]. The growing use of sporting events is an attempt to expand economic development opportunities and achieve tourism growth [6].

Management and planning issues are a key focus [7,8] and researchers are interested in the impacts events have on the host community [9]. Therefore, the impacts of sporting events on destination are many. The triple bottom line (TBL) is arguably the most widely accepted approach to identifying and measuring impacts [10], which is an assessment of economic, socio-cultural and environmental influences as pertinent to sporting events and sport tourism on the local communities [11–15].

Economic and related tourism benefits are seen as tangible or 'hard' impacts, and thus local stakeholders see the hosting of events as beneficial. Economic benefits include target investments in sport and event infrastructures, employment, prolonged tourism season, increased tourism, and new tax revenues [16–21]. Economic benefits also include some non-monetary effects, such as generating media attention and destination image enhancement [22–27]. One point of concern is the high (and sometimes excessive) costs and spending involved with building and preparing infrastructures, and this can result in increased taxes, higher prices and housing costs locally [28,29].

Social/cultural impacts are often considered 'soft' impacts [30,31], and scholars view the assessment, measurement and management of these impacts as more difficult [2,32]. Among these intangible impacts is a focus on local resident quality of life, enhanced social cohesion and pride in place, a new perceived destination image, and the potential to increase sport participation among locals [33–36]. However, a concern is that an increase in tourism can result in cultural conflicts among residents and tourists, traffic congestion problems, security and crime concerns, as well as vandalism—these are seen as negative impacts among local stakeholders observed in recent research [28,37,38].

Environmental impacts are also an important but challenging area of focus among scholars today. Some findings suggest that positive environmental impacts result when new sport infrastructures are built on devastated or reclaimed land and are a strategy to improve a site (see [39,40]), but in most cases local stakeholders perceive environmental impacts negatively. Without appropriate regulations and careful planning, new sports tourism infrastructure can lead to environmental consequences in a given area [28] and with high concentrations of people attending events, mass gatherings of event-goers see increases in waste, air and water pollution, as well as higher noise levels [28,29,41].

Although the main dimensions of sport event impacts are established (i.e., economic, socio-cultural and environmental), the scope of these dimensions are not unified—and sometimes result in very different impact outcomes. What is also noticed is that particular impacts could belong to different dimensions. Furthermore, some impacts are mentioned more often than others, leading to a conclusion that not all type of impacts are equally important, or at least not equally assessed. This makes this area of research open to further scrutiny and systematization. In addition, different stakeholders (e.g., organizers and managers, sponsors, spectators, active participants or locals) can have very different perspectives and impressions of an impact. Thus, an impact depends on many factors (type of the event, type of sport, demographics of the host community), but an accepted conclusion among researchers is that larger events result in greater impact, and these impacts can be positive and negative. In case of large-scale events, the impacts could spill over to non-host peripheral communities [40].

A challenge that researchers need to consider is that because each stakeholder group is involved differently in an event, they each have different expectations from an event, and thus value the importance of particular impacts of the event quite differently. While the event organizers and managers are actively involved in the event organization, the local population can also be actively (e.g., by volunteering or spectating) or passively involved into the event. However, considering the fact that locals live in the host destination, an event held in close geographical proximity will directly affect local residents. Their support plays an important role in making decisions about hosting and organizing the sporting event, the success of the event and further tourism development in the destination. It is therefore necessary to have a wide support and participation of the local community to ensure a long-term growth [42]. However, the extent of local resident support will depend on a balance between perceived benefits and costs of the event [40]. A critical point to consider is that not all members of the hosting community, such as local residents, local business organizations or financial institutions (see [43]) have an equal (positive or negative) perceptions of the impacts that the event they will host will have on their local community [44,45].

In general, event organizers want events to be sustainable, meaning that events will produce (in each dimension) more social benefits than costs to the overall community. It means that they seek, in collaboration with other stakeholders, to ensure financial viability and maximize other positive impacts, while eliminating or minimizing the negative ones in order to respect and appreciate the

interests of all those directly or indirectly involved and interested in the event itself. This certainly does not happen by chance, and sporting events need careful planning and implementation. In the planning stage, it is important to define specific (strategic) elements. This means defining event and destination attributes that will be a mandatory content to include when planning and preparing the event (e.g., ecological public traffic, recycling program, offer of local products, cultural program, employment of local population, event legacy), so that aspects can stay permanent over a prolonged period of time [6,46].

Consequently, in order to direct events on positive results and sustainability, for both the organizers and for the local population, it is necessary to focus on strategic elements necessary for efficient and effective event planning and organization. Indeed, decision-making, participation in events and loyalty to an event very often depend on satisfaction with various event elements/attributes in addition to entertainment, attractions, and supplemental destination activities [47–51]. Event practitioners have created a several important guides linking the need for sustainability principles within sporting event organization, including the Gold Framework [52]. While these guides provide just a framework, previous research has failed to study the relationship between specific strategic event and destination elements or attributes on one side, and event impacts on the other. If a relationship between a particular strategic element and a specific event impact exists, one can presume that putting more emphasis on that element during the planning stage will positively contribute to the realization of the related impact. In other words, there is little explanation of what strategic planning actions are necessary to undertake to produce specific positive impacts. This is especially the case with the staging of outdoor sporting events, which are more sensitive in ecological terms (see [53]).

This paper will address the wider impacts of sport-tourism events as well as strategic planning elements for achieving outdoor sport-tourism events' sustainability based on a systematic literature review of published journal articles. More precisely, it will review the published literature in order to systematize a wide spectrum of social impacts that outdoor sporting events create from the perspective of key stakeholders, as well as the strategic planning elements necessary for achieving sustainability. To accomplish this, we define the following research questions:

- RQ1: What is most commonly mentioned impact in sporting events relevant literature?
- RQ2: What types of sporting events are studied the most?
- RQ3: What strategic planning elements are necessary for achieving sustainability, based on what the literature assessed mentions as the most important?

This paper continues with a description of the methods used to conduct this systematic literature review before presenting the findings. The discussion and conclusion highlight key results to identify directions for future sport-event tourism studies.

## 2. Methodology

A systematic literature review method has often been applied to better understand diverse kinds of topics on sports tourism and sustainability-related research directions [54,55]. In addition, examining databases is an appropriate approach to exploring the extant of literature on the focus areas. The chosen database for this research was the Web of Science Core Collection (WoSCC) because this database includes top-rated peer-reviewed journals with international scope and coverage. While there are several complementary approaches to contribute a formal and systematic literature review [55–63], given that the study of sporting event organization and impacts is indeed heterogenic and involves multiple disciplines, methodological approaches and topics, this study used a two-phase process.

### 2.1. Phase 1

In conducting the review of impacts that sporting events create and referring to research question RQ1 and RQ2, we applied a systematic quantitative literature review. This answers one or more research questions, selects criteria for the inclusion/exclusion of the studies, requires a preliminary

overview of the current state of research, helps identifying research gaps and proposes future research agenda [55,56,62]. This particular study used a structured method to address particular research questions, thus aiming for methodological transparency and reliability, as well as systematic and comprehensive search on a specified topic. Moreover, we used a quasi-statistical approach in order to categorize, quantify, and identify trends in research over determined timeframes.

A comprehensive review was conducted from November 2019 to January 2020, thoroughly searching the WoSCC database for scientific journal papers published in the English language that contain the terms 'sport tourism event' and 'impact' in the topic (from titles, abstracts or keywords) from early publishing dates to the end of December 2019. While being aware of conceptual differences between the terms "event impacts" and "event legacies" [62,64], these two terms are often used interchangeably, and we therefore decided to search for both terms separately. Hence, another review was conducted in parallel, thoroughly searching WoSCC database for scientific journal papers published in the English language that contain the terms 'sport tourism event' and 'legacy'. These searches revealed 296 papers written in English language, from which 220 on "event impacts" research and 76 on "event legacies" research in the WoSCC database. However, an initial analysis of selected papers revealed that many of the identified publications were not research papers and/or dealt with the sporting event impact and legacy concept in an unsubstantial way. Based on the exclusion criteria, the final sample contains in total 77 papers.

When reviewing the final sample, the authors defined several categories (e.g., type of event, country, type of impact, type of element/attribute) by which to sort and quantify the data. Direct extraction helped classify data into categories, thus allowing new findings to emerge (see [62]). The first author stored, coded and categorized data manually and then counted frequency of appearance within particular categories. The first author shared preliminary results with the other two authors, and then all three authors jointly evaluated and agreed on the results.

## 2.2. Phase 2

When it comes to RQ3, that is, papers focused on strategic planning elements/attributes necessary for achieving sustainable events, we repeated a similar procedure as in Stage 1. The WoSCC database was first searched for scientific journal papers in English language that contain the terms 'sport', 'tourism', 'event' and 'element' in the topic (from titles, abstracts or keywords) also from early publishing dates to the end of December 2019. In parallel, the WoSCC database was searched for scientific journal papers in the English language that contain the terms 'sport', 'tourism', 'event' and 'element' or 'attribute' in the topic within the same period. There were considerably fewer papers and the two samples contained 33 papers in the research of 'sport, tourism, event, element' and 34 papers in the research of 'sport, tourism, event, attribute'. Among the publications, there were papers that referred to event and destination elements/attributes as determinants of tourists' satisfaction, experience, or revisit intention, and we excluded these publications from the analysis. Only a few of them referred to event and destination elements and/or attributes in relation to events' impacts and sustainability. Seven papers referred to strategic event and destination elements, and only ten referred to strategic event and destination attributes, making a final sample of 17 papers.

For more profound insight into the focal papers, authors applied content analysis. Content analysis allows researchers to analyze text systematically and to discover underlying concepts and hidden qualities and relationships between concepts [55]. The first author stored and categorized data manually regarding whether they are event or destination elements/attributes. Then, the inductive interpretation method (i.e., inductive coding) was useful to classify data into meaningful planning and organizational dimensions (see [62]). Therefore, the coded dimensions were derived straight from the text data. The first author shared their preliminary results with the other two authors, and then all three authors jointly evaluated and agreed on the results.

## 3. Results

### 3.1. Results of the Phase 1

The first research on events impacts and legacy contained 77 scientific papers written in the English language and all published in the 21st century (demonstrated in Figure 1). Fewer papers appear from 2002 to 2011, with increases shown from 2012 to 2019.

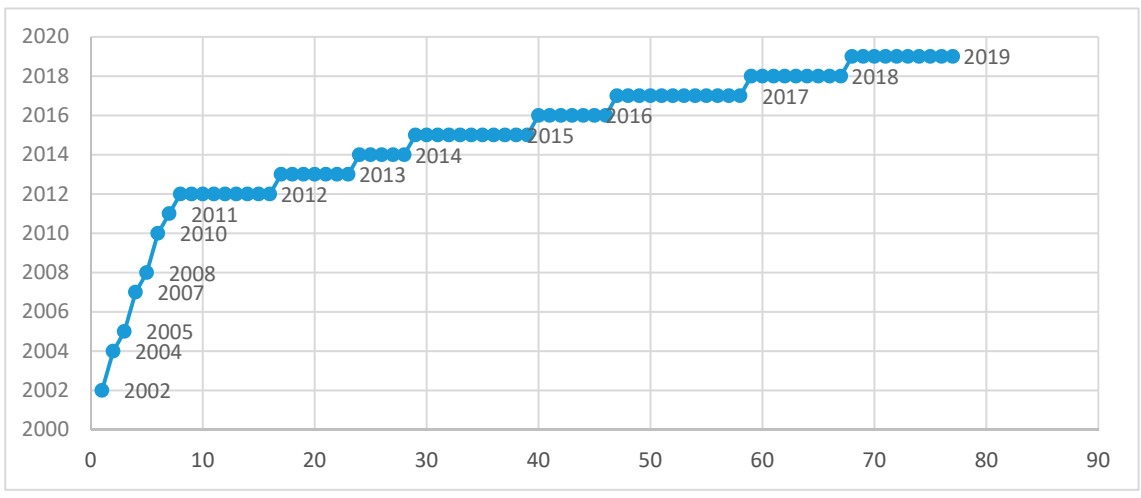

**Figure 1.** Number of published articles (2000–2019).

As displayed in Table 1 the journals with the most papers include *Sustainability* (eight publications) and *Tourism Management* (seven publications), followed by *European Sport Management Quarterly* (six publications) and *Journal of Sport Management* (five publications). It is evident, from the results, that these journals belong to the domains of sport, tourism, events, and sustainability.

**Table 1.** Most frequent sources/journals.

| Name of the Journal | Number of Papers |
|---|---|
| *Sustainability* | 8 |
| *Tourism Management* | 7 |
| *European Sport Management Quarterly* | 6 |
| *Journal of Sport Management* | 5 |
| *Sport Management Review* | 4 |
| *International Journal of Sport Policy and Politics* | 3 |
| *Current Issues in Tourism* | 2 |
| *Tourism Management Perspectives* | 2 |
| *Journal of Convention and Event Tourism* | 2 |
| *Leisure Studies* | 2 |
| Others | 36 |
| Total | 77 |

Among them, 60 research papers were empirical, 15 theoretical (literature reviews) and two included both theoretical and empirical research. There were a number of empirical studies conducted within different sports and sport events contexts. Most researched events were the Olympic Games: Summer Olympic Games (ten publications), Winter Olympic Games (eight publications) and Paralympic Games (one publication). World cups were also a common event focus, FIFA World Cups in particular with ten publications and other international and national races. Furthermore, most studies were focused on football or soccer (13 publications), cycle racing (four publications), running races, athletics, golf and Formula 1 (two publications), and rugby, skating, swimming and surfing with only one

publication. The rest relate to multi-sport events like the Olympic Games (18 publications). In empirical research studies, the authors collected data from key stakeholders (including, local residents, organizers, and participants), using surveys or interviews.

As shown in Table 2, the review reveals that the continent where the bigger number of studies were conducted is Europe (19 publications), followed by Asia (16 publications). A smaller number of studies were conducted in Africa (nine publications), mostly in the period of FIFA World Cups, and North and South America (eight and five publications, respectively). Concerning Australia, there were two studies. Three studies were conducted on different continents at the same time [65–67], with 15 conducted as a review of the existing literature without addressing any location.

**Table 2.** Dispersion of analyzed papers across continents.

| Continent | Number of Papers |
|---|---|
| Europa | 19 |
| Asia | 16 |
| Africa | 9 |
| North America | 8 |
| South America | 5 |
| Australia | 2 |
| Multiple | 3 |
| Others (no location) | 15 |
| TOTAL | 77 |

Table 3 displays the different countries most commonly studied. In the first row, South Africa and Korea (five publications), following with China, Croatia and USA (four publications) and Brazil and UK (three publications). Canada, Greece, Japan, Portugal were also research locations (two publications). There were four studies conducted in different countries at the same time [17,65–67]. Studies included in this paper show research conducted in more than 22 other nations across all inhabited continents.

**Table 3.** Dispersion of analyzed papers by country.

| Country | Number of Papers |
|---|---|
| South Africa | 5 |
| South Korea | 5 |
| China | 4 |
| Croatia | 4 |
| USA | 4 |
| Brazil | 3 |
| UK | 3 |
| Canada | 2 |
| Greece | 2 |
| Japan | 2 |
| Portugal | 2 |
| Multiple | 4 |
| Other countries | 22 |
| Other (no location) | 15 |
| Total | 77 |

As mentioned in the introduction, the organization and implementation of sport-tourism events create different economic, socio-cultural and environmental impacts on the local community (both positively and negatively). Table 4 systematizes the frequency of the main impact dimensions (economic, social and environmental impacts) and sub-dimensions from the research sample, as well as their categorization as being positive or negative impacts on/for the local community. Just to note, each paper usually referred to more than one impact (sub)dimension; so, this explains why the

total sum of all (sub)dimensions is higher than the number of analyzed empirical research papers (60 + 2 = 62 papers).

**Table 4.** Impacts and other sport-tourism event dimensions.

| Main Dimensions | | | Sub Dimensions | |
|---|---|---|---|---|
| | Frequency | | | Frequency |
| Economic | Positive | 81 | Tourism development | 25 |
| | | | Investments and benefits | 18 |
| | | | Destination image | 13 |
| | | | Sport development | 13 |
| | | | Business opportunities | 6 |
| | | | Job increase opportunities | 4 |
| | | | Economic sustainability | 2 |
| | Negative | 20 | Excessive investments | 14 |
| | | | Higher living costs | 5 |
| | | | Lack of strategic planning | 1 |
| Socio-cultural | Positive | 37 | Social heritage/capital | 13 |
| | | | Education and information | 7 |
| | | | Socio-cultural exchange | 6 |
| | | | Pride | 4 |
| | | | Increased interest in sports | 3 |
| | | | Political | 2 |
| | | | Psychological | 1 |
| | | | Social sustainability | 1 |
| | Negative | 9 | Disorder and conflicts | 6 |
| | | | Crime and vandalism | 3 |
| Environmental | Positive | 21 | Infrastructure and urban development | 15 |
| | | | Territory attractiveness | 4 |
| | | | Transportation | 2 |
| | | | Green technologies and resources | 1 |
| | Negative | 11 | Traffic problems | 7 |
| | | | Infrastructure | 3 |
| | | | Destroy of natural environment | 1 |

Scholars apply economic impact studies to a wide range of sporting and recreational events, which authors increasingly view them as an economic development tool (in addition to the achieving of recreation benefits). As displayed in Table 4, this research confirms the claim of previous studies that economic impacts are often more researched and the non-economic outcomes, especially environmental, are often overshadowed [42,68]. The results indicate 81 positive and 20 negative dimensions of economic impacts of sport-tourism events, 37 positive and 9 negative dimensions of socio-cultural impacts, and 21 positive and 11 negative dimensions of environmental impacts.

The positive economic impacts of sport-tourism events mostly focused on fostering tourism development, such as increased tourism revenues, development of tourism resources and infrastructure and tourism promotion. The results relate to other economic benefits such as the enhancement of destination/country image, increased investments, public infrastructure improvement, increased business and export opportunities for local companies, new job opportunities, and economic sustainability. Negative economic impacts mostly refer to the local population: higher living costs, higher prices of housing, goods and other local services, non-refundable investments with incorrect use of public funds, or lack of strategic planning. Positive socio-cultural impacts focused on community benefits, such as social sustainability contribution to residents lives, socio-cultural heritage and capital development, increased socio-cultural exchange, increased interest in sports and events, increased participation in sport/physical activity, community-oriented regeneration, engagement in social, cultural and educational event leverage, preservation of local traditions, and political and psychological benefits (e.g., increased pride and community spirit). The most often negative cited socio-cultural impacts were

crime and vandalism, terrorism, cultural conflicts, and lack of security. Positive environmental impacts involve infrastructure and urban development, attractiveness, transportation and green technologies concepts. Negative environmental impacts would mostly refer to traffic problems, infrastructure and the destruction of the natural environment.

Further, event host destinations were researched in 60 papers, non-host destinations only in one (i.e., [37]), and both host and non-host destinations in one (i.e., [69]). The other 15 literature review papers do not belong to either category (with data in these studies coming from existing literature).

### 3.2. Results of the Phase 2

In addition to the perceptions of residents of the host and non-host communities about the event impacts, this paper focuses on the strategic planning elements and attributes necessary for planning and organizing sustainable sporting events. There are 17 publications originating from this research that refer to strategic event and/or destination elements and strategic event and/or destination attributes. Figure 2 shows publications between 2009 and 2019, and indicates that research on both planning elements and attributes is a product of the last decade.

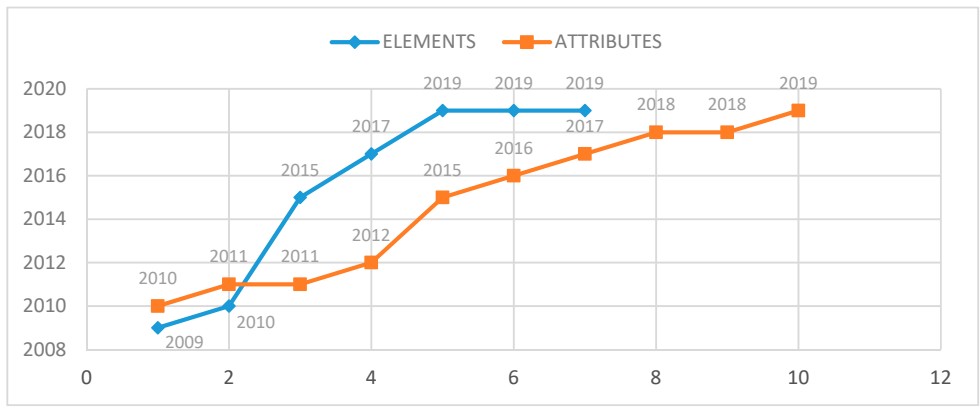

**Figure 2.** Number of published articles–event and destination elements/attributes (2000–2019).

Table 5 displays the results of the seven analyzed papers that refer to event and destination strategic and planning elements, while Table 6 presents the results of the ten analyzed papers that refer to event and destination strategic and planning attributes.

**Table 5.** Elements of sport-tourism events found in papers.

| Author(s) | Event Elements | Destination Elements |
|---|---|---|
| Perić, Vitezić and Đurkin Badurina [70] | Resources, Processes, Value Network | Infrastructure, Value Network |
| Kersulić and Perić [71] | Promotion, Preparation, Communication, Coordination | Promotion, Image |
| Aicher and Newland [72] | - | Trip Price, Leisure, Group Tours, Location, Shopping |
| Perić, Wise and Dragičević [73] | Participant Interaction, Organizational Processes, Resources, Sport Activity, Sport And Event Services | Environment, Tourism Services |
| Dong and Duysters [74] | Resources, Infrastructure, Facilities, Environment | Economic Development, Promotion Branding, Sport Culture |
| Ziakas [75] | Marketing Efforts, Entertainment, Coordination, Sport And Event Services | Promotion, Nature, Facilities, Tourism Services, Locals Support |
| Pettersson and Getz [76] | Location, Time, Interrelationships | - |

**Table 6.** Attributes of sport-tourism events found in papers.

| Author(s) | Event Elements | Destination Elements |
|---|---|---|
| Kruger and Viljoen [77] | Promotion, Support, Economy contribution, Tourists flow, Event services | Nature, Tradition, Tourism services, Leisure facilities, Friendliness of locals, Destination accessibility, Community Development |
| Su, Hsu, Huang, Chang [78] | Revisit intention | Nature, Facility, Human resources |
| Custodio, Azevedo and Perna [79] | Social problems, Traffic, Cultural development, Leisure | Economic development, Price level, Image, Environment |
| Lyu and Han [80] | Transportation, Sport services, Entertainment, Product price | Image, Massive tourism arrivals, Tourism service |
| Khor, Radzliyana and Lim [81] | Financial costs, Event services, Support, Entertainment, | Infrastructure, Tourism services, Self-achievement, Escapism, |
| Du, Jordan and Funk [82] | Support, Event location, Entertainment, Sport and event services | Image |
| Kaplanidou, Jordan, Funk and Ridinger [83] | Event location, Sport services | Image, Tourism services, Safety, Infrastructure and transportation, Shops, Nature, Leisure facilities, Sport facilities, Friendliness of locals, Destination accessibility |
| Agrusa, Kim and Lema [84] | Promotion, Exposure, Sport and event services, Resources, Infrastructure, Low costs | Tourism opportunities and services, Cultural attractions, Shops, Nature, Transportation, Inexpensive prices, Low crime and political canter |
| Hallmann and Breuer [85] | Sport and event services, Infrastructure | Image, Politics, Leisure facilities, Culture, Nature, Destination location, Infrastructure |
| Mohan [86] | Sport services | Natural resources, Infrastructure, Friendliness of locals, Politics, Costs, Tourism services, Safety, Leisure facilities, Shops, Culture |

The findings on researched strategic planning elements and attributes are multiple and are therefore matched and systematized based on eight dimensions, namely social, cultural, organizational, logistic, communication, economic, tourism, environmental factors (Table 7). The displayed dimensions are products of event and destination elements and attributes, systematized following previous research and analyzed articles.

**Table 7.** Systematized dimensions of event and destination strategic elements/attributes.

| Dimensions | Strategic Event Elements/Attributes | Strategic Destination Elements/Attributes |
|---|---|---|
| Cultural | Cultural development | Sport culture, Cultural attractions |
| Communication | Promotion, Popularity, Exposure | Promotion, Image, Branding |
| Economic | Prices, Costs, Economy contribution | Prices, Costs, Economic development |
| Environmental | Location/destination | Nature, Location/destination |
| Logistic | Resources, Infrastructure, Facilities | Infrastructure, Services, Facilities, Traffic |
| Organizational | Coordination, Preparation, Supporters, Entertainment, Services | Management, Organization |
| Social | Involvement, Interrelationship, Safety | Hospitality, Support, Safety, Politics |
| Tourism | New tourist markets, Services | Tourist opportunities, Services |

## 4. Discussion and Research Directions

Confirming the fact that research conducted on this subject relates to the more recent past, only a few research publications on impacts appear between 2002 and 2011, with most papers published 2012 to present. When taking into account the strategic planning elements and attributes together, those two terms belong primarily to research published in the second decade of the 21st century. While being aware that there is surely a number of papers on these topics published in the last century (but not included in the WoSCC database), these results demonstrate the novelty of the subject and mentioned terms in the research area, which suggests a lack of literature in the field of sport-tourism and events. It is important to note the increasing popularity of sport-tourism event research, which has

high research potential, especially with a focus on economic development across different geographical scales—regionally, nationally and globally.

When considering event host locations, regarding the sample of papers in this study, Europe (18 publications) and Asia (17 publications) are most represented. If taking North and South America as one, there are 13 publications altogether. Africa (nine publications) and Australia (three publications) saw fewer studies. The reasons why Africa and Australia lag behind in the academic literature is because these two continents do not host large-scale sporting events as often as Europe and Asia, and this can be due to the influence of recent mega-event hosting in emerging economy countries such as China and Russia, especially in recent years [87]. It can be because of the lack of existing and available large-scale sporting infrastructure on the Africa continent or the distance of the Australian continent given its geographical remoteness. Planners and organizers seek to improve the security level and prevent disorders and conflicts, choosing secure locations, especially when considering major sport events that require high-tech and modern infrastructures, which can be targets for causing alarm/disorders and even for crime and vandalism [2,38]. In addition, as participants and visitors who attend major sport events come from all over the world, venues have to be accessible and it is often preferred that host cities can be easily reachable from different parts of the world—so locations which are more distant rarely enter the final bidding stages.

When speaking about the different countries, South Africa and South Korea each appear in five publications. Next, a number of papers refer to China and USA, as high-economy center locations, but also, and surprisingly, Croatia (four publications). This result for South Africa contradicts previous conclusions about fewer studies focusing on Africa, but the reason why South Africa and Croatia were often studied lies in the fact that these destinations are economically stabile and popular tourist destinations (given the substantial amount of revenue that comes from tourism). Wise [88] added that sport research on Africa primarily focuses on South Africa. While this does not mean that South Africa or Croatia host many events (like large countries such as USA, the UK or China, who do host more large-scale events), when these countries host events, they are intriguing cases for academics to study event or tourism impacts. Other countries, like Brazil and the UK (three publications), Canada, Greece, Japan, and Portugal (two publications) were also identified as popular touristic destinations, and in that sense, are readily equipped with existing infrastructures, accommodations and hospitality-service providers necessary for event organizing and hosting to cater to participants and visitors. In fact, an important focus is attracting and retaining visitors to capture subsequent visitor spend before, during and after event, and stakeholders who will invest in certain business ventures are crucial for creating economic benefits for any country [16,18]. Synergistic impacts of hosting sporting events and touristic development, such as increases in tourist figures, seasonality fluctuations, new jobs, increased revenues and tax revenues from expenses were proven in previous studies [17,20,21,89–91]. Additionally, the impacts of sporting tourism will depend on the achieved level of development, numbers of sport tourists, size of local community, level of development and investment in other tourist activities, and if the local community accepts the hosting of sports tourism and sporting events. If the local community is underdeveloped (like in the case of Croatia), more products and services will be imported from other local communities or from abroad to provide service to the sport tourist, which would reduce the economic benefits for that place/country [92].

The results on sporting event impacts confirmed previous assertions that most of the focus was on positive as well as economic impacts. The results show that economic impacts mostly include benefits, such as new investment in infrastructures, new employment opportunities, increased tourism figures, and new tax revenues [16,17,20,21,89,93]. Another impacts that is considered a non-monetary effect is destination image enhancement [23,25]. Alternatively, planning for and hosting sporting events results in excessive spending (on the event), increased taxes locally, and higher living costs for local residents [28,29,94] as investors seek higher return yields. However, a new broad economic impact suggests economic sustainability has a positive perception. When compared to other impact types, the number of economic impacts identified in the sample confirms that they are more visible,

easier to measure and perceived as more beneficial than other types of impact which are somehow ignored [32,42].

Socio-cultural impacts have become a focus of attention mainly in the last few years. This research discusses the importance of non-monetary impacts. Studies have focused on sport participation, quality of life, social cohesion, formation of social capital, pride in place, and new or increased interest in a foreign country's culture and visitor attractions (see [33–36,94–97]). Each of these points on socio-cultural impacts can all lead to or enhance social sustainability. Given the short duration of the project with accent on economics impacts and necessities of long-terms policy initiatives, it is very important to direct future research to the benefits of socio-cultural impacts on the wellbeing of the community and residents.

Regarding environmental impacts, the results suggest that sporting events produce more positive than negative environmental impacts. This is opposite to many previous findings that highlight only/mainly negative environmental impacts, or where new sport events are a threat to the host community's environment. Considerations of threats found by studies include increased traffic problems and air pollution, high quantities of garbage, and higher noise levels (see [28,41,95,98,99]. Enhancing new infrastructure and dedicating urban regeneration/development strategies that focus on territory attractiveness and green technologies, resources and transportation are among the most often mentioned positive environmental impacts.

Negative economic and socio-cultural impacts of both small and large sporting events were not a key focus of study, and when assessed, they are always together with positive impacts outlined above and observed in recent studies [28,29,98,100,101]. A common conclusion is that academics prefer to emphasize positive event impacts and that they are not so keen to examine negative impacts of any type (economic, socio-cultural and environmental). As claimed by Getz and Page [2], Giampiccoli, Lee and Nauright [102] and Müller [103], opportunistic and/or political motives often lie behind this pattern of behavior, which presents an intriguing issue for future studies.

Furthermore, the fact is that the hos t destination is under the spotlight. The host community, first of all event organizers as well as local residents, is one of the most important stakeholders of an organized event. Without the support of locals, the organization of events will be difficult, especially in case of sport-tourism events which strongly depend on local public institutions, local companies and local volunteers [42]. The aforementioned studies mainly examined the attitude of local people concerning how sporting events in the host town will affect them and their communities [104,105].

Larger-scale events are not possible without the support of the wider community—that is, the non-host community. This offers an ambience where the social interaction between the visitors and local population facilitates and develops a positive image of the destination [106,107]. The communities that are not hosts can generate some benefits from sporting events especially for two reasons. First, the initial finance investment of larger events in non-host destination is minimal, in regards to the host destination. Secondly, the non-host destinations can dedicate themselves to the utilization of all resources because the event organization and management is not limited [95]. This research found only two existing studies that focused on the impacts of sporting events on non-host communities [37,69]. Still, previous studies noted differences in perception between host residents and non-host residents based on the proximity of the town to the event [108]. However, local residents of non-host towns are usually not questioned about the impacts of sporting event on their town (the so-called spill over impact) but about the impacts for the host town [95]. Therefore, the way that residents of places not hosting can estimate the impacts of sporting event on the actual host destination and its residents is still under-researched, and this need considered in future work going forward.

When looking at the context of sustainable tourism, outdoor sporting events must be sustainable, taking into account the above-mentioned impact dimensions on stakeholders. The legacies need advance planning from the early stages of event design and development. Legacy planning of the event and aspects which remain permanent, or which take some determinate period as well as human health and well-being and education, are factors that can aid organizational success. Therefore, except

for the impacts, specific strategic elements and attributes of sustainable sport-tourism events become very important characteristic when determining sustainability surrounding the organization of outdoor sport-tourism events. In order to focus on event elements and attributes, as the results of this research suggest, we identify several strategic elements/attributes and grouped them into eight distinctive categories, namely organizational, logistic, social, cultural, economic, tourism, environmental and communication dimensions.

Economic dimensions focus on costs and benefits (as suggested by Perić et al. [70] and Aicher and Newland [72]), while tourism dimensions refer to new or complementary markets and increasing the number of tourists to events and in destinations. Social dimensions involve the hospitality of local residents, safety and political issues (see Kaplanidou et al. [83] and Mohan [86]) as well as participant preferences. Cultural dimensions relate to local culture, art and historic attractions, sport culture and cultural development. Environmental dimensions are also important (see for instance [79,85]), and reveal geographical location, natural setting and attractions (especially clean and green environment) and traffic elements/attributes, as each important. All of this was coordinated from the event and destination, for instance organizational and logistical dimensions (elements/attributes such as management, coordination, infrastructure, resources, facilities, entertainment, strategy and monitoring) and the communication dimension (which puts focus on promotion, image, branding, identity and popularity as exposure in mass media). These destination and event strategic elements and attributes are important for designing strategies for planning and development of future events (see for instance [73,75]). In other words, all these dimensions can guide strategies to help better understand different stakeholder needs, improve event organization, and leverage event legacies. As the 17 publications about strategic organizational elements/attributes did not address the direct relationship with events impacts, the direction for future research suggests the need to analyze those relationships: between event strategy and/or destination elements/attributes (or dimensions) and specific event impacts.

Indeed, societal needs, shared values and meeting individual group needs, equal rights/access are important points of focus. As suggested by Zhang and Park [6], legacy planning and aspects that last, human health and education, policy and principles, monitoring and evaluation processes, and reduced consumption of natural resources are the most important factors (from the perspective of the multiple event stakeholders) when considering sustainable development and responsible hallmark event planning and operations. Comparing this study's results with factors suggested by Zhang and Park [6], organizational, logistic and communication dimensions can be considered legacy planning and aspects that last, monitoring and evaluation processes and policy and principles while social and cultural dimensions can be considered as human health and education. Zhang and Park [6] concluded that these dimensions are even more important for sustainable event organizing. Economic and tourism factors are new dimensions necessary for planning and achieving business legacies, better organization, and preventing risk. Special emphasis is given to the environmental dimension. Global concern now suggests the need to focus on event impacts alongside biodiversity and the environment, and urban renewal and biodiversity. Therefore, some considerations for event planners focusing on environmental impacts include being conscious of any ongoing ecosystem regeneration and dedicated preservation areas, as these are protected areas and events can harm such efforts. These are important to understand when implementing and preparing for future planning and organization of sport-tourism events. For this reason, the category named the environmental dimension is proposed. This category matches the reduced consumption of natural resources category that relates to development over time [6]. Those factors, and not only, should be used in planning for future sustainable sporting events. It is crucial to use the findings of previous studies in combination with good practices as a base to contribute to the given theoretical framework. In terms of practice, events should have a dedicated cultural program. To achieve this, it is important that the local residents are involved with the event and participate in the event(s), and (perhaps most importantly) jobs are created for local residents (and these jobs need to be sustained, opposed to seasonal or temporary).



## 5. Limitations and Concluding Remarks

Limitations do exist when conducting systematic quantitative literature reviews. The first limitation relates to the defining of the research boundaries for systematic review. This involves transparent inclusion and exclusion of research papers. The key words (i.e., 'sport tourism event' and 'impacts'; 'sport tourism event' and 'legacy'; 'sport' and 'tourism' and 'event' and 'element'; 'sport' and 'tourism' and 'event' and 'attribute') in key fields (from titles, abstracts or keywords) means we have maybe excluded some articles that were still of relevance (conceptually). This means that some papers that might discuss the idea of 'impact' and 'legacy' but which use different terminology (i.e., outcome, consequence, benefit, develop and transform) have been excluded from our sample. This problem is a common occurrence in literature review papers (see [61]). In addition, the criteria for only evaluating academic peer-reviewed journal articles using WoSCC mean that some influential studies that appear in monographs, book chapters and postgraduate theses were not a part of this analysis. Moreover, the requirement for articles published in English in this study can result in limited insight, but this is a common approach with such studies to limit the focus to one publication language or event specific journals [60]. These limitations are largely due to the scope and focus of the research framework agreed when preparing the analysis for this paper, but this does position and frame future research opportunities to compare work published in other languages.

To conclude, this paper contributes to the relevant theories of sports management, event management, tourism management and stakeholder management. The systemization and categorization of event impacts from the perspective of key stakeholders and strategic event and destination elements/attributes are necessary for successful and sustainable event organization. Moreover, the proposed eight strategic dimensions can also act as a framework for future studies on event organization and event impacts. Furthermore, this paper has several practical implications. Foremost, discovering key stakeholder perceptions of impacts and elements/attributes of sustainable outdoor sport-tourism events is useful for organizers of future sport-tourism events to consider. This is important when trying to get the support of the entire community and to gain a better understanding how residents experience both positive and negative event impacts. In addition, by determining the elements and attributes of sport-tourism events, which are crucial for achieving sustainability, this can help planners and organizers. If someone perceives one type of impact as the most important, in the preparation phase, that individual will then consider relevant elements/attributes of events—this will contribute to the realization of a particular impact. Indeed, without the inputs from stakeholders, the planning of outdoor sport-tourism events could be much harder. The research results could facilitate organizers and help them to avoid mistakes when planning, organizing, financing and managing future events with a particular emphasis on sustainability principles. Knowledge and experience in organization of sustainable sport-tourism events is crucial, contributing not only to the sustainable development of tourism, but also to the prolonged economic sustainability.

**Author Contributions:** A.C.T.: paper concept, data collection, and writing of analysis and conclusions. M.P.: paper concept and writing of conclusions. N.W.: support for concept and writing of conclusions. All authors have read and agreed to the published version of the manuscript.

**Funding:** This work has been fully supported by the University of Rijeka under the project number uniri-drustv-18-103.

**Conflicts of Interest:** The authors declare no conflict of interest. The funders had no role in the design of the study; in the collection, analyses, or interpretation of data; in the writing of the manuscript; or in the decision to publish the results.

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
