# Peer review of "Assessing and Considering the Wider Impacts of Sport-Tourism Events: A Research Agenda Review of Sustainability and Strategic Planning Elements"

_sustainability, doi:10.3390/su12114473_

Round 1

Reviewer 1 Report

Formal aspects
First, there are several recent papers, including those published in this journal, have tackled other areas of Sustainable Tourism, sport Tourism, etc. I recommend that you review these papers and similar, as they provide a clearer presentation of the literature.
• Garrigos-Simon et al. (2018); Tourism and sustainability...
• Gómez-Trujillo, A.M.; Vélez-Ocampo, J.; González-Pérez, M.A. Una revisión de la literatura sobre la causalidad entre la sostenibilidad y la reputación corporativa. ¿Qué va primero? Manag. Reinar. Qual. 2020, 31, 406-430. https://doi.org/10.1108/MEQ-09-2019-0207
• Jiménez-Garcia, M,*, Ruiz-Chico, J., Peña-Sánchez, A.R. and López-Sánchez, J.A. (2020); A Bibliometric Analysis of Sports Tourism and Sustainability (2002–2019). Sustainability, 12(7),2840; https://doi.org/10.3390/su12072840
• Linnenluecke et al. Conducting systematic reviews and bibliometrics analyses. 2019
• Niñerola et al. (2019); Tourism research on sustainability. Sustainability.
• Serrano et al. (2019); Using Bibliometric Methods to Shed Light on the Concept of Sustainable Tourism. Sustainability.
Second, the bibliography is not in alphabetical order. There are also poorly cited articles (Please modify it). Title of Site web available online: URL (accessed on Day, Month and Year).
Third, the tables and figures are not homogeneously cited.

Conceptual aspects
The title does not reflect the content of the article.
There are statements that must be explained, (paragraphs 58 and 59) for example; “Environmental impacts could be positive when new sport infrastructure is built on devastated land (see [39,40], but in most cases, local stakeholders perceive environmental impacts as negative…” you need to hit the target from the start and the organization should be:
I. Why/how sport tourism got to this point.
II. Despite all the positives aspects of Sport tourism (economic, cultural, social) sustainability is an issue.
III. Due to the problems, we need answers on tourism and sustainability and that requires research.
IV. How does good planning influence sustainability?
V. Methods for researching it; this is where you explain bibliometry vs systematic reviews and meta-analyses. We need this info to understand why bibliometry is the right choice.
I think you should use VOSviewers software and not do it manually to be more effective. This software is a widely used tool to process keywords and eliminate coincidence or synonyms. On the other hand, the SciMAT software is used for this type of analysis because it facilitates the organization of topics in categories (motor grouping, highly developed and isolated grouping, emerging or decreasing, and basic and transversal grouping) depending on the centrality and density indicators according to Callon's model. It's a lot to do, but it’s realistic and would make this paper a solid work to guide future researchers.

Author Response

Response to Review Report Form 1

Authors Comments and Responses

We would like to thank the reviewer from the Review Report No 1 for all the recommendations, suggestions and comments.

First, following the reviewer’s comments, we made some changes to explain better the parts which were not clear. We also made some changes on English language spell check which was required, as recommended by reviewers.

The literature (i.e. references) as well as the format of the tables and figures are according to the instructions for authors.

Second, referring to the Formal aspects part, following reviewers’ statements, we included few additional references (those suggested plus an additional) which they find were relevant to the article, to provide a clearer presentation of the literature, methodology and to complete the article.

Third, in line with reviewers’ comments of the Conceptual aspects part, we made some changes on the title of the article to reflect better the content of the paper. In addition, we tried to explain better the statements in the mentioned paragraphs.

Continuing with the Conceptual aspects part, we believe our paper covers all the mentioned aspects. We would also like to thank reviewer for beneficial suggestions regarding methodology. However, being aware of many types of literature review and many tools (software) how to do it, we find more appropriate the use of the systematic quantitative review for the research on events impacts. Systematic quantitative review is recognized and accepted in academic practice (Pickering and Byrne, 2014; Cheng et al, 2018; Thomson et al., 2019). Regarding elements/attributes and their dimensions, the content analysis approach was used. The Methodology and Results sections are revised accordingly. However, we agree on using the proposed method/software in our future research as this review paper begins to show directions from the published literature.

We thank you for your comments on this review paper for the special issue on Urban and Rural Event Tourism and Sustainability: Exploring Economic, Social and Environmental Impacts.

22 May 2020

Reviewer 2 Report

The objective of the paper is to perform a bibliometric analysis on the impact of sport-tourism events. The subject is interesting and well justified. However, the work presents several shortfalls that should be solved.

First, the authors conduct two different searches from which they draw their conclusions. The reason for the two searches and the relationship between them are not justified. The selection of the terms is not clear: I tried to replicate the search but I only got 10 documents. Authors should offer a more accurate description of this aspect. Moreover, the analysis of the two groups of articles is not clear. The explanation of the second group seems to start in the middle of a paragraph (lines 242-243), without a clear relationship with the previous text.

Second, the tools they use to perform the analysis should be improved. It is unclear whether the authors would like to perform a bibliometric analysis or a content analysis. In the first case, it would be advisable to use some specific software and bibliometric indicators (citations, co-citations, co-occurrences, density, etc.), to gain a better understanding of the patterns of knowledge.

In the case of content analysis, the analysed works should be further explored. It would be interesting to analyse aspects such as: types of works, techniques of analysis, samples used, units of analysis, etc.

On the other hand, it is not clear how the topics have been obtained or how they have been grouped. In particular, Table 7 should be explained. The meanings of the concepts "elements" and "attributes" are also unclear, which adds confusion about the objective of the work and the contribution of the results.

Third, the conclusions do not clearly show the contribution of the manuscript or how it comes from the analysis developed. This section should be better organized, clearly justifying the descriptive results on the one hand, and the gaps found, as well as future lines, on the other.

Finally, authors should revise writing. There are some paragraphs very similar to paragraphs from other authors (see, for example, lines 39-62, 313-316, 421-435).

Author Response

Response to Review Report Form 2

Authors Comments and Responses

First, we would like to thank the reviewer for all the recommendations, suggestions and comments. We are also very glad reviewer found this study interesting and well justified.

Following the reviewer’s comments, we made some changes to explain better the parts which were not clear. We also made some changes on English language spell check which was required, as recommended by reviewers.

In line with reviewers’ recommendations, we clarified the objective(s) of the paper: to review and systematize a wide spectrum of social impacts that outdoor sport-tourism events create from the perspective of key stakeholders and addresses strategic planning elements necessary for achieving event sustainability.

Following the reviewer’s suggestions, we tried to explain better the relationship between the research on impacts (legacy) and planning elements/attributes needed for achieving these impacts. We also explained better (in the Introduction) the meanings of the concepts "elements" and "attributes" which are often used simultaneously.

We also explained better the applied methodology. Our intention was not to present bibliometric indicators (citations, co-citations ...). Instead, we opted for the systematic quantitative review for the research on events impacts. Systematic quantitative review is recognized and accepted in academic practice (Pickering and Byrne, 2014; Cheng et al, 2018; Thomson et al., 2019). We covered many of the mentioned aspects (empirical/conceptual papers, journals, continents/countries, type of sports, host/non-host communities, etc.) you mentioned. Regarding elements/attributes and their dimensions, the content analysis approach was used. We explained in more details how the common dimensions have been obtained and how they have been grouped (in Table 7). The selection of the search terms is explained too. Therefore, we split the methodology in two parts (Phase 1 and Phase 2). Accordingly, the results of the two phases are presented separately too.

At the end, authors tried to better organize and clearly show the contribution of the manuscript. In the case of review paper, the descriptive results are justified and the gaps and lines for future results are going to identify the needs and some directions for works and key areas of focus.

Finally, to avoid similarities to paragraphs from other authors, as recommended, authors revised writing as per the highlights from MDPI report.

We thank you for your comments and points to improve this review paper.

22 May 2020

Round 2

Reviewer 1 Report

ready to postreaready to post

Reviewer 2 Report

The present manuscript shows more clearly the methodology, the purpose of the study and how this study is developed.